# Organoids Are Limited in Modeling the Colon Adenoma–Carcinoma Sequence

**DOI:** 10.3390/cells10030488

**Published:** 2021-02-25

**Authors:** Yoshihisa Tokumaru, Masanori Oshi, Ankit Patel, Wanqing Tian, Li Yan, Nobuhisa Matsuhashi, Manabu Futamura, Kazuhiro Yoshida, Kazuaki Takabe

**Affiliations:** 1Department of Surgical Oncology, Roswell Park Comprehensive Cancer Center, Buffalo, NY 14263, USA; yoshitoku1090@gmail.com (Y.T.); masanori.oshi@roswellpark.org (M.O.); ankit.patel@roswellpark.org (A.P.); 2Department of Surgical Oncology, Graduate School of Medicine, Gifu University, 1-1 Yanagido, Gifu 501-1194, Japan; nobuhisa@gifu-u.ac.jp (N.M.); mfutamur@gifu-u.ac.jp (M.F.); kyoshida@gifu-u.ac.jp (K.Y.); 3Department of Gastroenterological Surgery, Yokohama City University Graduate School of Medicine, Yokohama 236-0004, Japan; 4Department of Biostatistics & Bioinformatics, Roswell Park Comprehensive Cancer Center, Buffalo, NY 14263, USA; Wanqing.Tian@RoswellPark.org (W.T.); li.yan@roswellpark.org (L.Y.); 5Department of Surgery, Niigata University Graduate School of Medical and Dental Sciences, Niigata 951-8510, Japan; 6Department of Surgery, University at Buffalo Jacobs School of Medicine and Biomedical Sciences, The State University of New York, Buffalo, NY 14263, USA; 7Department of Breast Oncology and Surgery, Tokyo Medical University, 6-7-1 Nishishinjuku, Shinjuku, Tokyo 160-8402, Japan; 8Department of Breast Surgery, Fukushima Medical University School of Medicine, Fukushima 960-1295, Japan

**Keywords:** organoid, colorectal cancer, adenoma–carcinoma sequence, tumor microenvironment, GSEA, xCell, CIBERSORT

## Abstract

The colon adenoma–carcinoma sequence is a multistep genomic-altering process that occurs during colorectal cancer (CRC) carcinogenesis. Organoids are now commonly used to model both non-cancerous and cancerous tissue. This study aims to investigate how well organoids mimic tissues in the adenoma–carcinoma sequence by comparing their transcriptomes. A total of 234 tissue samples (48 adenomas and 186 CRC) and 60 organoid samples (15 adenomas and 45 CRC) were analyzed. We found that cell-proliferation-related gene sets were consistently enriched in both CRC tissues and organoids compared to adenoma tissues and organoids by gene set enrichment analysis (GSEA). None of the known pathways in the colon adenoma–carcinoma sequence were consistently enriched in CRC organoids. There was no enrichment of the tumor microenvironment-related gene sets in CRC organoids. CRC tissues enriched immune-response-related gene sets, whereas CRC organoids did not. The proportions of infiltrating immune cells were different between tissues and organoids, whereas there was no difference between cancer and adenoma organoids. The amounts of cancer stem cells and progenitor cells were not different between CRC and adenoma organoids, whereas a difference was noted between CRC and adenoma tissues. In conclusion, we demonstrated that organoids model only part of the adenoma–carcinoma sequence and should be used with caution after considering their limitations.

## 1. Introduction

Colorectal cancer (CRC) is the third most diagnosed cancer worldwide [1]. Despite significant advancement in multimodal treatment options during recent years, CRC remains the second leading cause of cancer-related deaths, trailing only lung cancer [1,2]. To this end, there remains a need for novel therapeutic agents to treat this disease.

Pre-clinical models using cancer cell lines have been the main vehicles for drug development [3]. The advantage of this model is in the simplicity and reproducibility its offers in testing and screening drugs. However, the monoclonal cancer cells used in this model are unable to accurately reflect cancer biology or the tumor microenvironment (TME). In order to overcome this limitation, animal models are used [4,5,6]. However, syngeneic models only provide data on rodent cancer [5,6], while xenograft models, including patient-derived xenografts (PDX), ignore immune response [7]. The lack of an appropriate system to model human cancer is the reasons many phase III clinical trials fail [8]. Recently, cancer organoids have been used in pre-clinical studies for their superior efficiency in performing drug sensitivity tests as compared to PDX models [4]. PDX models can take 6–8 months before an investigation can be completed, whereas tumor organoids only require 1–3 months [4,9]. It is thought that cancer organoids mimic human cancer biology with higher fidelity than cancer cell lines, but it remains unclear how closely cancer organoids resemble human cancers and their TME [4].

The majority of CRCs are caused by progressive genomic alterations referred to as the adenoma–carcinoma sequence [10,11]. This sequence is usually initiated with a genetic alteration of the *APC* gene that allows formation on an adenoma, which is subsequently followed by the alterations of the *KRAS* and *TP53* genes [11]. The dysregulation of the pathways related to cell proliferation and apoptosis, including the WNT, RAS-MAPK, PI3K, TGF-β, and TP53 pathways, progresses to the carcinogenesis of CRC [12]. Matano et al. utilized organoid models to recapitulate this sequence and analyzed the significance of each pathway involved [12]. Although the morphology, mutational status, and drug response of cancer organoids have been extensively studied [13,14,15], comprehensive transcriptomic analyses that compare the adenoma–carcinoma sequence in tissues and organoids has not been performed. To this end, we aimed to clarify whether organoid models mimic the adenoma–carcinoma sequence in fresh-frozen tissue samples obtained from two cohorts from the Gene Expression Omnibus (GEO) database—GSE41258 (adenoma and CRC tissue cohort) [16] and GSE57965 (adenoma and CRC organoid cohort) [12].

Recently, our group has performed in silico translational research to identify biomarkers [17,18,19], clinically relevant immune cells [20,21,22,23], predictive genes [24], as well as microRNAs in breast and gastrointestinal cancers [25]. We employed a computational algorithm, referred to as gene set enrichment analysis (GSEA), which enables us to analyze the differences in biological pathways between two distinct groups [26]. We utilized CIBERSORTx [27] and xCell [28], which uses transcriptomic data to estimate the composition of immune cells within human tumors. In this study, we hypothesized that mechanisms of the adenoma–carcinoma sequence are maintained in organoid models, and sought to clarify the utility of organoid models for future investigations.

## 2. Materials and Methods

### 2.1. Data Acquisition of Colorectal Cancer Cohorts

The transcriptomic and clinical data for colorectal cancer (CRC) and adenoma were obtained from three cohorts—one cohort of tissues (GSE41258) [16] and two cohorts of organoids (GSE57965 [12] and GSE74843) [29]. GSE41258 holds a total of 390 tissue samples, which include 186 primary CRCs, 48 adenomas (polyps), 1 high grade polyp, 47 liver metastases, 20 lung metastases, 2 microadenomas, 54 normal colons, 13 normal livers, 7 normal lungs, and 12 cell lines. Among these, we utilized a total of 234 (48 adenoma and 186 primary CRC) tissue samples. GSE57965 holds a total of 23 samples, which include 5 colon adenoma organoids, 3 colon epithelial organoids, 7 CRC organoids, 6 genome-engineered organoids, and 2 metastatic CRC organoids. Among these, we utilized a total of 12 (5 adenoma organoids and 7 CRC organoids) samples in the current study. GSE57965 holds a total of 58 samples, which include 10 adenoma organoids, 38 CRC organoids, 7 normal colorectal mucosa organoids, and 3 serrated organoids. Among these, we utilized a total of 48 samples (10 adenoma organoids and 38 CRC organoids). In an attempt to verify our results, we identified two other studies that analyzed the transcriptome of organoids. The study by van de Wetering et al. (GSE64392) [14] includes only CRC organoids and normal tissue organoids, but does not include adenoma organoids. Thus, it was unusable for our analyses. The study data in the study by Weeber et al. [13] was not deposited to the Gene Expression Omnibus, and therefore we were unable to access it. Given that all of the cohorts used in the current study are from a publicly accessible and deidentified database, ethics approval was waived by the Institutional Review Board.

### 2.2. Gene Set Enrichment Analysis (GSEA) and Single-Sample GSEA

The Broad Institute provided the publicly available software, gene set enrichment analysis (GSEA) (http://software.broadinstitute.org/gsea/index.jsp) [26]. The Broad Institute was founded by Massachusetts Institute of Technology and Harvard University (https://www.broadinstitute.org/) in 2004. In the current study, we utilized hallmark gene sets of Molecular Signature Database collections, as we have reported previously [17,18,25,30]. The statistical significance of the false discovery rate (FDR) was set to 0.25, as recommended by the GSEA developer Broad Institute. We also performed single-sample GSEA to demonstrate the absolute enrichment value of each sample by calculating an enrichment score for the gene set of interest [31].

### 2.3. CIBERSORTx

CIBERSORTx, the algorithm that enabled us to analyze the proportions of 22 immune cells within the tumor microenvironment of tissue samples, was developed and reported by Newman et al. [27].

### 2.4. Statistical Analysis

To perform the statistical analysis, we utilized the publicly available software R (version 4.02). The statistical significance was analyzed using Fisher’s exact test or one-way ANOVA. For the box plot figure, one-way ANOVA was performed. Median values are represented as black lines inside boxplots (Tukey type) and the spans of rectangles demonstrate interquartile ranges. The statistical significance was defined as a two-sided *p* value of less than 0.05.

## 3. Results

### 3.1. Cell-Proliferation-Related Gene Sets Were Enriched in Both Colorectal Cancer (CRC) Tissue and Organoid Cohorts

Cancer is more proliferative than benign adenoma tumors given the nature of malignancy. In order to investigate whether cancer organoids are more proliferative than adenoma organoids, we conducted gene set enrichment analysis (GSEA) of hallmark cell-proliferation-related gene sets on cohorts that included both adenoma and CRC transcriptomes from tissue (GSE41258, n = 48 and n = 186, respectively) and from organoids (GSE57965, n = 5 and n = 7, respectively). As expected, CRC tissues significantly enriched hallmark cell-proliferation-related gene sets, such as E2F target, G2M checkpoint, and mitotic spindle, but not Myc targets V1 or V2 (Figure 1A, Appendix A; normalized enrichment score (NES) and false discovery rate (FDR); NES = −1.47, FDR = 0.133; NES = −1.56, FDR = 0.121; NES = −1.60, FDR = 0.167; NES = −0.94, FDR = 0.560; NES = −1.09, FDR = 0.439, respectively). *MKI67*, the most frequently used cell proliferation marker in the clinical setting, was significantly elevated in CRC tissue compared to adenoma tissue (Figure 1B, *p* = 0.008).

Interestingly, CRC organoids enriched hallmark cell-proliferation-related gene sets more than CRC tissue. CRC organoids enriched E2F targets, G2M checkpoint, Myc targets V1 and V2, but not mitotic spindle (Figure 1C, Appendix A; NES = −1.58, FDR = 0.100; NES = −1.48, FDR = 0.132; NES = −1.52, FDR = 0.118; NES = −1.91, FDR = 0.002; NES = −1.09, FDR = 0.493, respectively). On the other hand, the expression levels of MKI67 were not statistically different between CRC and adenoma organoids, despite a difference in median value, likely because of the large variance and small sample size of CRC organoids (Figure 1D, *p* = 0.187). To overcome the sample size limitation, we performed a single-sample GSEA from the GSE57965 cohort. The results of single-sample GSEA were strikingly similar to that of larger cohort GSEA. The gene sets that were enriched in CRC tissue, such as E2F targets, G2M checkpoint, and Myc targets V1 and V2, were significantly enriched in CRC organoids as well. In addition, CRC organoids had a higher GSEA score in the single-sample GSEA than adenoma organoids (Figure 1E; *p* = 0.024, *p* = 0.037, *p* = 0.036, and *p* < 0.001 respectively). We additionally analyzed the GSE74843 cohort, which included adenoma organoids and CRC organoids. In this cohort, CRC organoids enriched the Myc target V2 gene set, which was further validated in the single-sample GSEA. CRC organoids demonstrated a higher single-sample GSEA score than adenoma organoids in the Myc target V2 gene set (Appendix A).

The noted enrichment of cell-proliferation-related gene sets suggests that cancer organoids may be as proliferative as adenoma organoids and that CRC tissue may be as proliferative as adenoma tissue.

### 3.2. Among the Pathways in the Colon Adenoma–Carcinoma Sequence, Only the MTORC1 Gene Set Was Enriched in CRC Organoids

Several pathways are known to contribute to the development of CRC from a benign adenoma, also known as colon adenoma–carcinoma sequence [11,12]. It was of interest to investigate whether the same pathways were enriched in tumor tissues and organoids. GSEA of those pathways was conducted on tissue (GSE41258) and organoid (GSE57965) cohorts. In the tissue cohort, CRC tissue enriched all of the pathways involved in the adenoma–carcinoma sequence, including WNT β-catenin, KRAS signaling, MTORC1, and TGF-β pathways (Figure 2A, Appendix A; NES = −1.36 FDR = 0.184; NES = −1.57 FDR = 0.136; NES = −1.58 FDR = 0.150; NES = −1.30 FDR = 0.234, respectively). On the other hand, only the MTORC1 pathway was enriched in CRC organoids in the GSE57965 cohort (Figure 2B, Appendix A; NES = −1.38 FDR = 0.197). The TGF-β pathway was enriched in adenoma organoids (Figure 2B, NES = −1.68, FDR = 0.152), which was consistent with the GSE74843 cohort (Figure 2C; NES = −1.77 FDR = 0.182).

These results suggest that CRC organoids do not enrich as many of the adenoma–carcinoma sequence-related gene sets as CRC tissues.

### 3.3. Only CRC Tissues Enriched Tumor Immune Microenvironment (TME)-Related Gene Sets and Correlated with a Higher Infiltration of Stromal Cells When Compared to Adenoma Tissues

Given the nature of organoids, it was of interest to investigate whether they maintained the signaling pathways and contained the infiltrating cells of the tissue TME. CRC tissue enriched TME-related gene sets, including angiogenesis, but not adipogenesis (Figure 3A, Appendix A; NES = −1.93 FDR = 0.024; NES = −0.86 FDR = 0.659, respectively). In contrast, CRC organoids did not enrich either of these gene sets (Figure 3B). CRC tissue analysis correlated with a higher infiltration of stromal cells in the TME as compared to adenoma tissues, including fibroblasts, lymphatic endothelial cells, microvascular endothelial cells, and pericytes (Figure 3C; *p* < 0.001, *p* < 0.001, *p* < 0.001, and *p* = 0.031). In addition, the stroma score was significantly higher in CRC tissues (Figure 3C; *p* < 0.001). In contrast, stromal cells were not highly infiltrated in CRC organoids and the corresponding stroma score was low (Figure 3D). In adenoma organoids from the GSE74843 cohort, only fibroblasts were highly infiltrated and the stroma score was significantly elevated (Appendix A). These results imply that both adenoma and CRC organoids have poorly developed TMEs, meaning there is no difference in cell infiltration or TME-related gene expression profiles.

### 3.4. CRC Tissues Enriched Immune-Response-Related Gene Sets but CRC Organoids Did Not

It was previously demonstrated that highly proliferative cancer is associated with an enhanced immune response in the TME [32]. As expected, CRC tissue enriched immune-response-related gene sets, such as allograft rejection, interferon (IFN)-α response, IFN-γ response, inflammatory response, IL-6/JAK/STAT3, complement, and coagulation (Figure 4A, Appendix A; NES = −1.43 FDR = 0.145; NES = −1.42 FDR = 0.150; NES = −1.48 FDR = 0.135; NES = −1.55 FDR = 0.119; NES = −1.44 FDR = 0.144; NES = −1.46 FDR = 0.138; NES = −1.52 FDR = 0.130, respectively). CRC organoids and adenoma organoids did not enrich any of the immune-response-related gene sets in either cohort (GSE57965 (Figure 4B) or GSE74843 (Appendix A)). These results suggest that immune activity in CRC organoids is not different from immune activity in adenoma organoids.

### 3.5. The Proportions of Infiltrating Immune Cells Are Different between Tissue and Organoid Cohorts

Utilizing the CIBERSORT algorithm, the cell compositions of immune cell types were estimated from each transcriptomic profile. In adenoma tissues, nearly 60% of the cell population was composed of lymphocytes, half of which were B cells (Figure 5A). Among the myeloid cells, monocytes composed 29.8% of the population. In CRC tissues, nearly 75% of cells were lymphocytes. B cells were the most abundant overall cell type (Figure 5A) and granulocytes were the most abundant myeloid cell type in CRC tissues.

In adenoma organoids, the proportion of lymphocytes was similar to that in tissue samples, but the most abundant cell type was T cells (Figure 5B). Of the myeloid cells, granulocytes were the most abundant cell type in adenoma organoids. A similar trend was observed in CRC organoids, which contained a large proportion of lymphocytes, but the most abundant cell type was T cells. Regarding myeloid cells, the population of granulocytes was abundant in CRC organoids as well.

### 3.6. B Cells and T Cells Were Less Infiltrated and Macrophages Were More Infiltrated in CRC Tissues Compared to Adenoma Tissues, Whereas There Were No Differences in Organoids

The proportional differences of immune cells prompted us to analyze the cell composition differences of lymphocytes and myeloid cells between adenomas, CRC tissues, and organoids. Among T cells, infiltration of CD4 memory resting T cells and regulatory T cells (Tregs) was low, whereas follicular T cells were higher in CRC tissue (Figure 6A; *p* < 0.001, *p* = 0.0018, and *p* = 0.002, respectively). Only CD4 memory resting T cells were significantly lower in CRC organoids compared to adenoma organoids in both GSE57965 (Figure 6A; *p* = 0.012) and GSE74843 cohorts (Appendix A; *p* = 0.046). CD4-memory-activated cells were significantly higher in only the GSE57965 cohort (Figure 6A; *p* = 0.049). All types of macrophages (M0, M1, and M2) were highly infiltrated, whereas activated dendritic cells (DC) were less infiltrated in CRC tissue (Figure 6B; macrophages all *p* < 0.001, DC *p* = 0.046). CRC organoids demonstrated consistently higher infiltration of M0 cells in the GSE74843 cohort, whereas activated DCs (Appendix A; *p* = 0.015) were consistently infiltrated in CRC organoids from the GSE57965 cohort (Figure 6B; *p* = 0.005). The infiltration of memory B cells and plasma cells was low in CRC tissue compared to adenoma tissue, but there was no difference in B cell infiltration between CRC and adenoma organoids (Appendix A). There was no difference in infiltration of NK cells and eosinophils between CRC tissue, adenoma tissue, or organoids (Appendix A). Resting mast cells demonstrated higher infiltration in CRC organoids, but activated mast cells demonstrated higher infiltration in adenoma organoids from the GSE74843 cohort (Appendix A). Neutrophils were more prevalent in CRC tissues, but not in CRC organoids (Appendix A).

### 3.7. There Was No Difference in the Infiltration of Stem Cells and Progenitor Cells between Adenoma and CRC Organoids

Given the nature of malignancy, we expected CRC tissues to have high infiltration of mesenchymal stem cells and progenitor cells. As expected, CRC tissues were associated with high infiltration of mesenchymal stem cells (Figure 7A; *p* < 0.001), but there was no difference in hematopoietic stem cell infiltration. In contrast, CRC organoids demonstrated no difference in either type of stem cell when compared to adenoma organoids in both GSE57965 and GSE74843 cohorts (Figure 7B, Appendix A). Unexpectedly, CRC tissues had less infiltration of multipotent progenitor cells and common myeloid progenitor cells, but had greater infiltration of megakaryocyte–erythroid progenitor cells (Figure 7C; *p* < 0.001, *p* < 0.001, and *p* < 0.001, respectively). Only megakaryocyte–erythroid progenitor cells were highly infiltrated in CRC organoids from the GSE74843 cohort (Appendix A). There was no difference in progenitor cells between CRC and adenoma organoids in the GSE57965 cohort (Figure 7D). These results imply that both CRC and adenoma organoids possess stem-cell-like characteristics, and thus there is no difference in stemness between the organoid types.

## 4. Discussion

In the current study, we found that cell-proliferation-related gene sets were enriched in both colorectal cancer (CRC) tissues and organoids. Only the MTORC1 pathway, among all of the pathways involved in the adenoma–carcinoma sequence (WNT β-catenin, KRAS signaling up, MTORC1, and TGF-β pathways), was enriched in CRC organoids of the GSE 57965 cohort. In comparison, all of the pathways were enriched in CRC tissue. CRC tissue enriched angiogenesis and infiltrated stromal cell gene sets, represented by fibroblast and vascular endothelial cell infiltration, whereas CRC organoids did not. CRC tissue enriched immune-response-related gene sets, which was not seen for CRC organoids. The ratio of immune cells was different between tissues and organoids. Lymphocytes were more abundant in CRC tissue compared to adenoma tissue. B cells and T cells were less infiltrated and macrophages were highly infiltrated in CRC tissue. There was no difference in lymphocyte or myeloid cell infiltration between CRC and adenoma organoids. There were no differences in cancer stem cells or progenitor cells between CRC and adenoma organoids. There was a difference in cancer stem and progenitor cells between CRC and adenoma tissue, which implies that organoids are inherently rich in these cells regardless of their derivation from cancer or adenoma.

Organoids are an important part of drug development and drug sensitivity screening, especially from the perspective of personalized medicine, given the ease with which they can be generated from patient tumors and the fidelity of their genomic profile to the original tumor. Organoids are more costly in comparison to cell line models, however they are reported to be more economical in comparison to PDX models [4]. As an advantage, organoids require less time to complete drug response assays compared to PDX models [9,14]. An expediated time frame could increase transability and support early clinical availability of drugs for clinical use.

CRC organoids enriched the cell-proliferation-related gene sets, which was concordant with CRC tissues. Multiple studies have investigated the gene expression alterations between tumors and their derivative organoids. Matano et al. reported that cancer organoids maintained the aggressive biology of cancer tissues by expressing CRC-specific genes [12]. Fujii et al. further demonstrated that CRC organoids preserved the same gene signatures and gene alterations of the original CRC tissue, as assessed by GSEA [29]. In agreement, Weeber et al. reported that cancer organoids demonstrated similar genetic profiles compared to the original tumor [13]. Furthermore, van de Wetering et al. reported the establishment of an organoid bank in which organoids possessed the genetic alterations representative of CRC, such as alterations in *APC*, *TP53*, *PIK3CA*, and *KRAS* [14]. In the current study, the transcriptome of the organoids was not compared to the tumor it was derived from. The original study by Matano et al. compared adenoma tissues and organoids to CRC tissues and organoids using GSEA to analyze the genomic fidelity of derived organoids to the original tumors [12]. The novelty of our approach is that we were able to study cancer biology as represented by the hallmarks of cancer progression (cell proliferation, inflammation, and metabolism) through the GSEA of hallmark gene sets of the Molecular Signature Database. We found that CRC organoids enriched the cell-proliferation-related gene sets more than adenoma organoids. The same relationship is present in the comparison between CRC tissues and adenoma tissues.

The mutational status and drug responses of organoids have been extensively studied [13,14,15]. However, this is the first study that has analyzed the biology of the adenoma–carcinoma sequence by comparing the transcriptomics of CRC and adenoma tissues and their respective organoids. We had expected that CRC organoids would enrich the gene sets related to the adenoma–carcinoma sequence. To our surprise, our analysis demonstrated the enrichment of only the MTORC1 pathway. These results may be due to the fact that GSEA is analyzing multiple genes instead of one certain gene.

A limitation of organoids is that they do not proportionally reflect the composition of the tumor microenvironment as represented by tissue-resident immune cells or vasculature promoting cells of the original cancer tissue [29,33]. The novelty of the current study is that it investigates the differences in stromal and vasculature formation between CRC organoids and adenoma organoids. The results indicated that there was lack of fidelity in the proportional loss of tissue-resident and vasculature-promoting cells between CRC and adenoma organoids versus tissues. This implies that immune and vascular cells are lost during the organoid development process, and thus no similarities were maintained.

In the current study, we found that the tumor immune microenvironment of the organoids is very different from that of tissues, and it does not reflect the changes of the adenoma–carcinoma sequence. This result suggests that interpretation of experimental results using organoids should be done with caution when analyzing immune cell functions. To overcome this issue, Neal et al. proposed a co-cultured system of cancer organoids and endogenous tumor infiltrating lymphocytes to test the drug response of anti-PD-1 or anti-PD-L1 immune check point inhibitors [33]. Outside of this artificial reconstitution of the TME, the complex interaction between organoids and immune cells cannot be reliably interpreted.

The infiltration of stem cells and progenitor cells, which are cells of multipotency and undifferentiation [34], did not differ between CRC and adenoma organoids, but differed between CRC and adenoma tissues. These results may reflect the ability of organoids to maintain stem cell characteristics, regardless of the tissue of origin (adenoma or cancer) [35,36]. Whether this stem-cell-rich environment of the organoid system interferes with its ability to model the original tumor tissue is to be determined with further investigation.

The current study has obvious limitations. One of the limitations stems from our inability to compare tissue samples and organoid samples in the same population. This was due to the lack of access to a cohort that has both tissue and organoid transcriptomic data. In addition, our results may not reflect the spatial difference within the bulk tumor and the heterogeneity among the patients due to the small number of organoid samples. Although we believe that the statistical significance is real when the difference persists despite the small sample size, we may not be encompassing many findings due to the small sample size of the organoids. Further investigation comparing organoid and tissue samples directly is warranted and would make our findings more intriguing and convincing.

In conclusion, we demonstrated that the organoids mimic only part of the adenoma–carcinoma sequence of colorectal cancer development. Thus, one needs to use organoids with caution due to their limitations.

## Figures and Tables

**Figure 1 cells-10-00488-f001:**
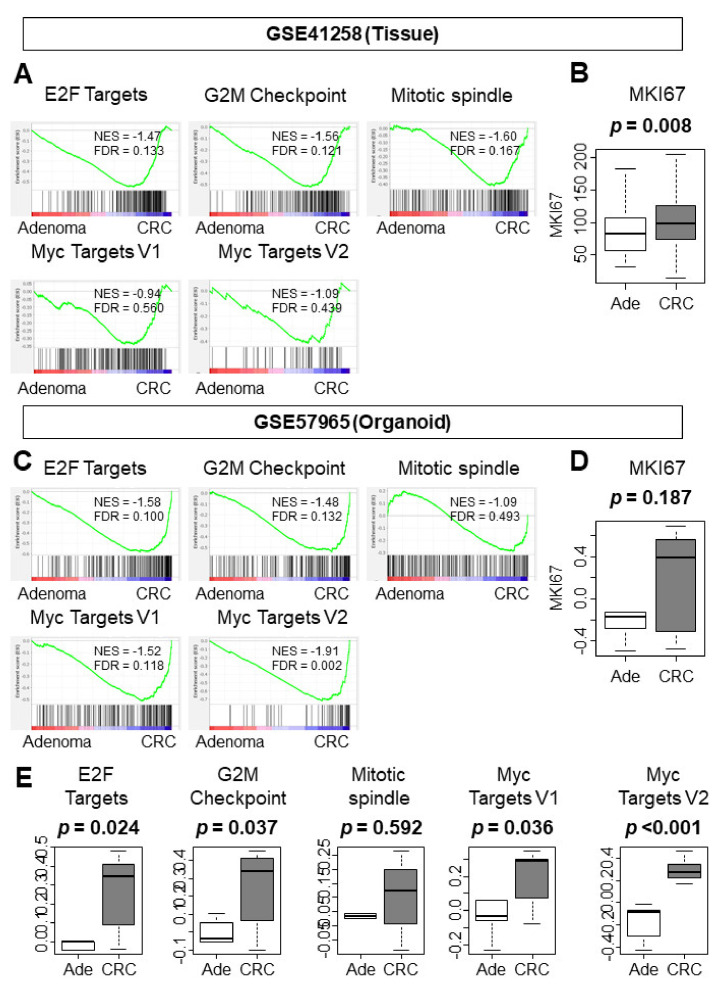
Gene set enrichment analysis (GSEA) of cell-proliferation-related gene sets and analysis of *MKI67* expression. (**A**) GSEA of adenoma vs. CRC tissue in GSE41258. (**B**) The MKI67 expression levels of adenoma and CRC in tissue. (**C**) GSEA of adenoma vs. CRC organoids in GSE57965. (**D**) The comparison of MKI67 expression levels between adenoma and CRC in organoids. Tukey-type boxplots demonstrate the median and interquartile level values. (**E**) Single-sample GSEA of GSE57965. Statistical significance was defined as a false discovery rate (FDR) < 0.25. Ade, adenoma; CRC, colorectal cancer; FDR, false discovery rate.

**Figure 2 cells-10-00488-f002:**
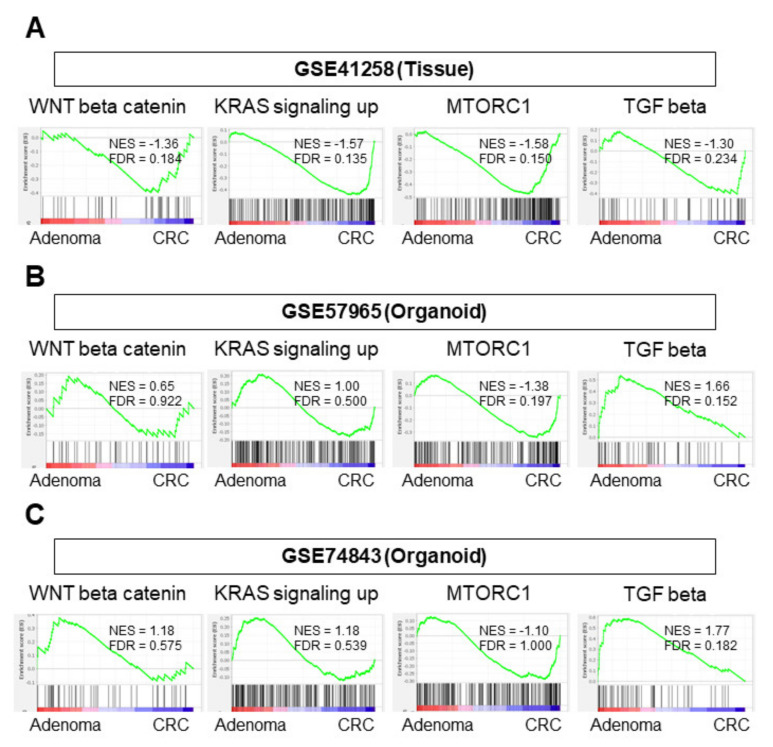
GSEA of gene sets associated with an adenoma–carcinoma sequence. (**A**) Analysis of tissue sample (GSE41258). (**B**) Analysis of organoid sample (GSE57965). (**C**) Analysis of organoid sample (GSE74843). Statistical significance was defined as a false discovery rate (FDR) < 0.25. CRC, colorectal cancer; FDR, false discovery rate; NES, normalized enrichment score.

**Figure 3 cells-10-00488-f003:**
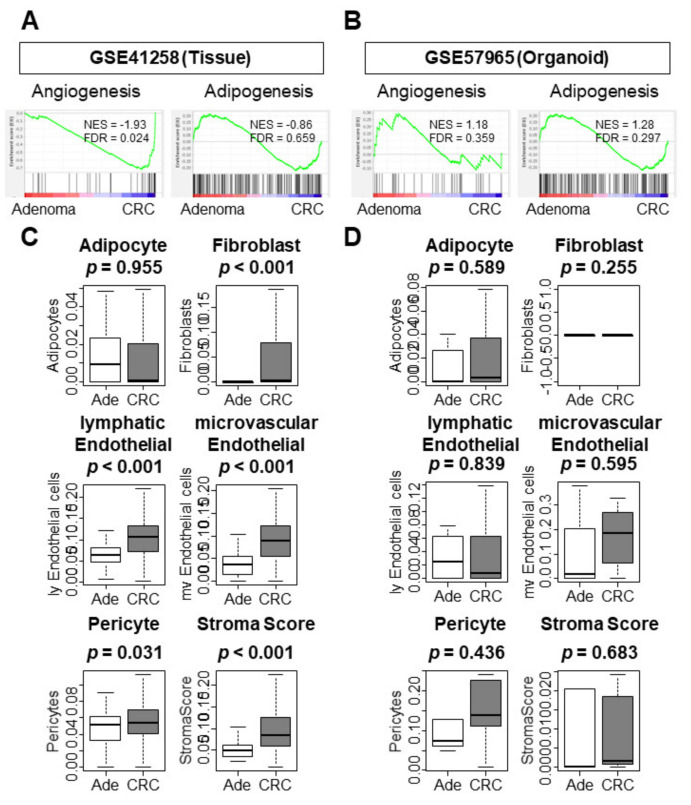
GSEA of tumor immune microenvironment (TME)-related gene sets, the infiltration of stromal cells, and comparison of stroma scores. (**A**) GSEA of tissue sample (GSE41258). (**B**) GSEA of organoid sample (GSE57965). (**C**) Infiltration of stromal cells and comparison of stroma scores in tissue samples (GSE41258). (**D**) Infiltration of stromal cells and comparison of stroma scores in organoid samples (GSE57965). Tukey-type boxplots demonstrate the median and interquartile level values. Statistical significance was defined as a false discovery rate (FDR) < 0.25. Ade, adenoma; CRC, colorectal cancer. NES, normalized enrichment score.

**Figure 4 cells-10-00488-f004:**
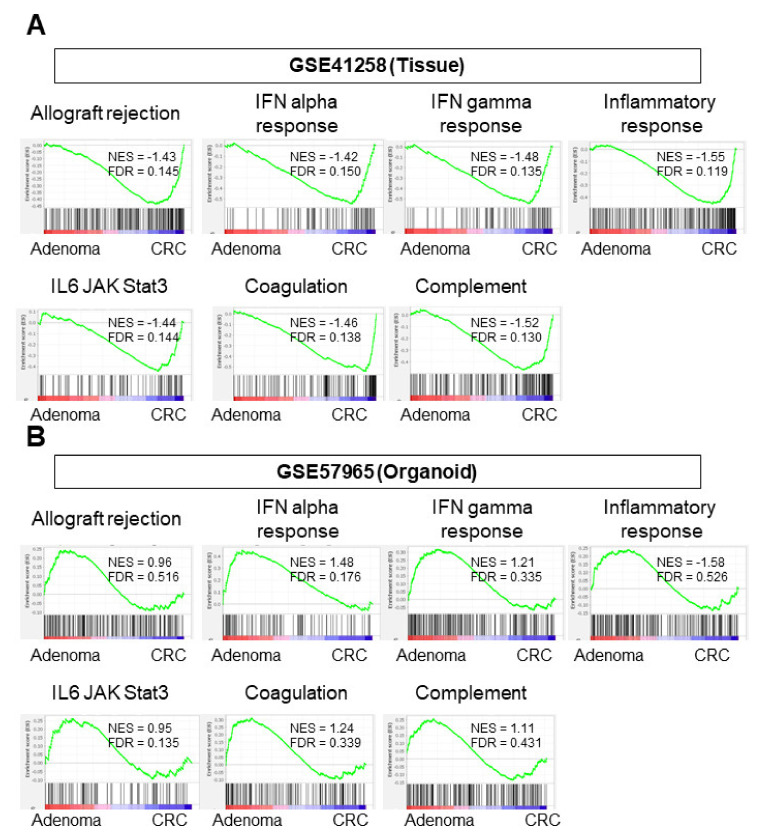
GSEA of gene sets associated with immune response. (**A**) Analysis of tissue sample (GSE41258). (**B**) Analysis of organoid sample (GSE57965). Statistical significance was defined as a false discovery rate (FDR) < 0.25. CRC, colorectal cancer; FDR, false discovery rate.

**Figure 5 cells-10-00488-f005:**
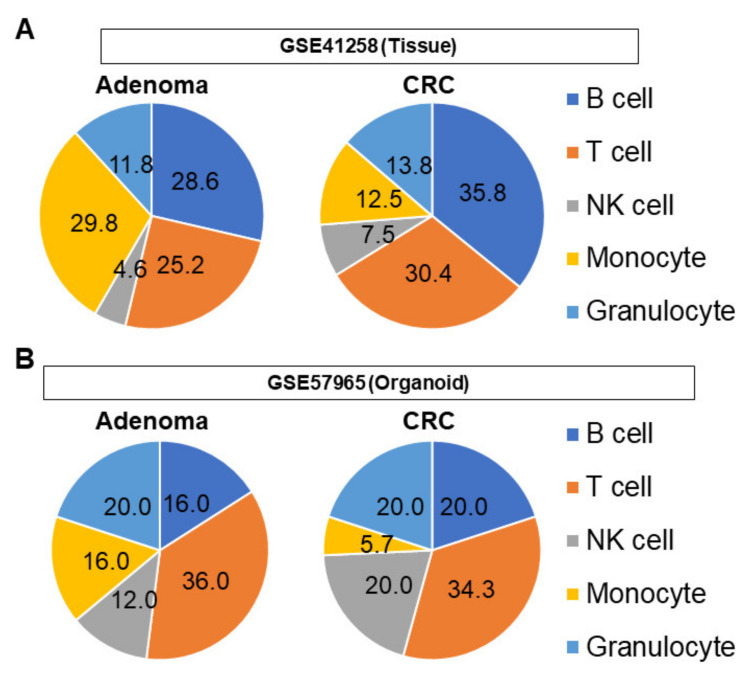
Pie chart demonstrating the proportion of infiltrating immune cells. (**A**) Analysis of tissue sample (GSE41258). (**B**) Analysis of organoid sample (GSE57965). NK cell, natural killer cell.

**Figure 6 cells-10-00488-f006:**
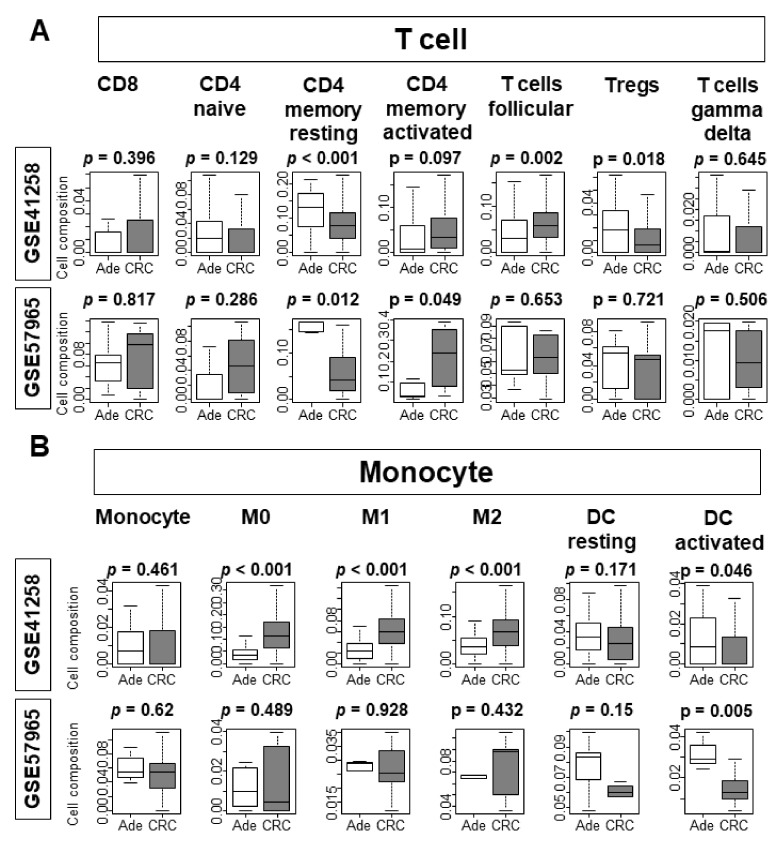
Comparison of immune cell infiltration. (**A**) Analysis of T cells. (**B**) Analysis of monocytes. Tukey-type boxplots demonstrate the median and interquartile level values. Ade, adenoma; CRC, colorectal cancer; Tregs, regulatory T cells; M0, Macrophage M0; M1, Macrophage M1; M2, Macrophage M2; DC, Dendritic Cell.

**Figure 7 cells-10-00488-f007:**
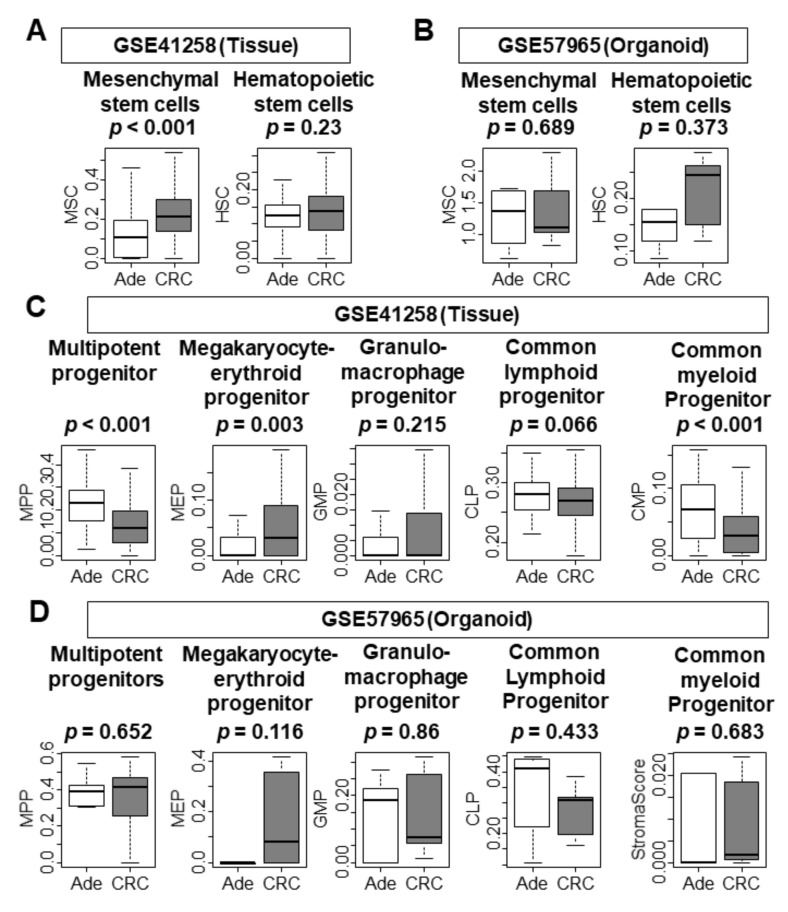
Comparison of stem cell and progenitor cell infiltration between adenoma and CRC tissues and organoids. (**A**) Infiltration of stem cells in tissue samples (GSE41258). (**B**) Infiltration of stem cells in organoid samples (GSE57965). (**C**) Infiltration of progenitor cells in tissue samples (GSE41258). (**D**) Infiltration of progenitor cells in organoid samples (GSE57965). Tukey-type boxplots demonstrate the median and interquartile level values. Statistical significance was defined as a false discovery rate (FDR) < 0.25. Ade, adenoma; CRC, colorectal cancer.

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
