# Peer review of "Organoids Are Limited in Modeling the Colon Adenoma–Carcinoma Sequence"

_cells, 2021, doi:10.3390/cells10030488_

Round 1
Reviewer 1 Report
The paper investigates the correspondence of the genomic alteration in adenoma-carcinoma sequence in colorectal cancer.
Transcriptomes of cancer and adenoma cells derived from both organoids and tissues were investigated in order to evaluate the reliability of results obtained from the in vitro experiments on organoids to assess drugs efficacy.
Some significant differences were observed in genomic alteration suggesting that the organoids mimic only part of the adenoma-carcinoma sequence and the need to use organoids with caution in in vitro investigations.
A short paragraph on the gene amplifications procedure reporting also data on the accreditation state of laboratories performing analysis (if available) may be useful to readers.
In figure 1 the Tukey-type boxplots of MKI 67 seem to be not consistent with the p values.
Author Response
Comment 1:
The paper investigates the correspondence of the genomic alteration in adenoma-carcinoma sequence in colorectal cancer. Transcriptomes of cancer and adenoma cells derived from both organoids and tissues were investigated in order to evaluate the reliability of results obtained from the in vitro experiments on organoids to assess drugs efficacy. Some significant differences were observed in genomic alteration suggesting that the organoids mimic only part of the adenoma-carcinoma sequence and the need to use organoids with caution in in vitro investigations.
Response 1:
First of all, we would like to thank Reviewer 1 for taking his/her time and putting his/her effort to review our manuscript. The constructive comments helped us improve our manuscript. We are very pleased that Reviewer 1 has noted the points that we wanted to emphasize in our manuscript.
Comment 2:
A short paragraph on the gene amplifications procedure reporting also data on the accreditation state of laboratories performing analysis (if available) may be useful to readers.
Response 2:
We agree with Reviewer 1 that it would be useful to the readers to add a short paragraph describing about Broad Institute which provides the gene set enrichment analysis (GSEA). Therefore, we added a sentence describing about Broad Institute as follows.
Broad Institute provides the publicly available software, Gene set enrichment analysis (GSEA) (http://software.broadinstitute.org/gsea/index.jsp) [26]. Broad Institute was founded by Massachusetts Institute of Technology and Harvard University (https://www.broadinstitute.org/) in 2004.
Comment 3:
In figure 1 the Tukey-type boxplots of MKI 67 seem to be not consistent with the p values.
Response 3:
We agree with Reviewer 1 that the boxplot of MKI67 (Figure 1D) seems to be inconsistent with the p value. There was no statistically significant difference in MKi67 expression between adenoma organoid and CRC organoid (p = 0.187) despite remarkable difference in median value. This is most likely because of a large variance due to a small sample size of organoids. This was added in the Result section (Line 142 - 145).
On the other hand, MKI67 expression level was not statistically different between cancer and adenoma organoids despite remarkable a difference in median value, likely because of a large variance due to a small sample size of CRC organoids (Figure 1D, p = 0.187).
Reviewer 2 Report
This is a well written article that shows an interesting work on the topic of cancer cell derived organoids, specially focused on adenoma to colorectal cancer (CRC) transition, which describes the result of transcriptomic analysis comparisons of these pathophysiological states between patient tissue samples and in vitro generated organoids from the two stages.
By using Gene Set Enrichment Analysis (GSEA) and the CIBERSORTx algorithm, the authors describe the differences between adenoma tissue and CRC tissue, and make a comparison with adenoma-derived organoids and CRC-derived organoids. Their results are sound and useful to take into account when working with organoids as a mean to screen pharmacological therapies in the context of personalized medicine as well as in the generation of models for the study of CRC.
However, I would like to raise two questions that could be of interest for potential readers:
Question 1:
According to GEO database, there are some discrepancies in the samples used to carry out the present transcriptomic analysis:
Dataset GSE57965 is composed by 23 samples of whom, 5 are adenoma-derived organoids and 9 are CRC-derived organoids. However, the authors say that they used 5 adenoma-derived organoids and only 7 CRC-derived organoids for their analysis.
Dataset GSE41258 is composed of 390 samples including primary CRC tumor, polyps, liver metastasis, lung metastasis, microadenoma, as well as normal colon, liver and lung tissues and several cell lines. Among them, there are 186 primary CRC tumors and 49 polyps (adenomas). However, the authors describe that they used 186 CRC samples and only 48 adenomas for their analysis.
Could the authors explain why not all the available samples for CRC-derived organoids and adenoma tissue were used in the analysis? In case that some reason exist for the selection of samples to be analysed, it should be clearly stated to allow reproducibility.
Question 2:
The main conclusion of the article is that there is a clear limitation of CRC organoids to reflect completely the features of a CRC tumor, especially regarding to adenoma to CRC transition and tumour microenvironment (immune cell infiltration, stem cell and progenitor presence). Could the authors comment on possible causes for this incomplete modelling provided by organoids? Differences in the proportions of infiltrated immune cells and stroma are maybe due to the necessary in vitro period for the establishment of the organoid, and stem cells and progenitors differences regarding adenoma or CRC tissue maybe due to in vitro selection forced by culture media and used conditions.
Author Response
Comment 1:
This is a well written article that shows an interesting work on the topic of cancer cell derived organoids, specially focused on adenoma to colorectal cancer (CRC) transition, which describes the result of transcriptomic analysis comparisons of these pathophysiological states between patient tissue samples and in vitro generated organoids from the two stages. By using Gene Set Enrichment Analysis (GSEA) and the CIBERSORTx algorithm, the authors describe the differences between adenoma tissue and CRC tissue, and make a comparison with adenoma-derived organoids and CRC-derived organoids. Their results are sound and useful to take into account when working with organoids as a mean to screen pharmacological therapies in the context of personalized medicine as well as in the generation of models for the study of CRC. However, I would like to raise two questions that could be of interest for potential readers:
Response 1:
We would like to thank Reviewer 2 for taking his/her time and putting his/her effort to review our manuscript. We are delighted to learn that Reviewer 2 found our results to be sound and useful, and are grateful to have comments that strengthen our manuscript.
Comment 2:
According to GEO database, there are some discrepancies in the samples used to carry out the present transcriptomic analysis:
Dataset GSE57965 is composed by 23 samples of whom, 5 are adenoma-derived organoids and 9 are CRC-derived organoids. However, the authors say that they used 5 adenoma-derived organoids and only 7 CRC-derived organoids for their analysis.
Dataset GSE41258 is composed of 390 samples including primary CRC tumor, polyps, liver metastasis, lung metastasis, microadenoma, as well as normal colon, liver and lung tissues and several cell lines. Among them, there are 186 primary CRC tumors and 49 polyps (adenomas). However, the authors describe that they used 186 CRC samples and only 48 adenomas for their analysis.
Could the authors explain why not all the available samples for CRC-derived organoids and adenoma tissue were used in the analysis? In case that some reason exist for the selection of samples to be analysed, it should be clearly stated to allow reproducibility.
Response 2:
We would like to thank Reviewer 2 for pointing out this very important issue that requires clarification. GSE57965 holds 9 colorectal cancer (CRC) organoids as Reviewer 2 has mentioned. However, only 7 out of 9 CRC organoids were derived from primary CRC, whereas remaining 2 were derived from metastatic CRC. Given that the biology of metastatic CRC can be very different from the primary CRC, we chose to analyze only 7 organoids derived from primary CRC for the current study.
GSE41258 holds 49 polyp (adenoma) tissues as Reviewer 2 has mentioned. Among the 49, 1 was defined as “high grade polyp” whereas the others were defined as “polyp”. In order to maintain uniformity, we excluded the high grade polyp and analyzed only 48 polyp tissues in the current study.
To make these points clear, we changed the sentences in the section “2.1 Data acquisition of colorectal cancer cohorts” as follows.
GSE41258 holds a total of 390 tissue samples which includes: 186 primary CRC, 48 adenomas (polyps), 1 high grade polyp, 47 liver metastases, 20 lung metastases, 2 microadenomas, 54 normal colons, 13 normal livers, 7 normal lungs, and 12 cell lines. Among those, we utilized a total of 234 (48 adenomas and 186 primary CRC) tissue samples. GSE57965 holds a total of 23 samples which includes: 5 colon adenoma organoids, 3 colon epithelial organoids, 7 CRC organoids, 6 genome-engineered organoids, and 2 meta-static CRC organoids. Among those, we utilized a total of 12 (5 adenoma organoids and 7 CRC organoids) samples in the current study.
Comment 3:
The main conclusion of the article is that there is a clear limitation of CRC organoids to reflect completely the features of a CRC tumor, especially regarding to adenoma to CRC transition and tumour microenvironment (immune cell infiltration, stem cell and progenitor presence). Could the authors comment on possible causes for this incomplete modelling provided by organoids? Differences in the proportions of infiltrated immune cells and stroma are maybe due to the necessary in vitro period for the establishment of the organoid, and stem cells and progenitors differences regarding adenoma or CRC tissue maybe due to in vitro selection forced by culture media and used conditions.
Response 3:
We agree with Reviewer 2 that additional comments on possible causes for incomplete modelling by organoids would be informative to the readers. Indeed, necessary in vitro period for the establishment of the organoid most likely contributed to differences in the proportions as well as the fewer number of infiltrated immune cells and stroma cells. Further, in vitro selection by culture media and used conditions most likely contributed to differences in stem cell and progenitor cell in tissue between adenoma and CRC. We added this comment in Discussion section as below (Line 327 - 334).
A limitation of organoids is that they do not proportionally reflect the composition of the tumor microenvironment as represented by tissue-resident immune cells or vasculature promoting cells of the original cancer tissue [31,32]. The novelty of the current study is that it investigates the difference in stromal and vasculature formation between CRC or-ganoids and adenoma organoids. The results indicated that there was lack of fidelity in the proportional loss of tissue-resident and vasculature promoting cells between CRC and adenoma organoids versus tissues. This implies that immune and vascular cells are lost during the organoid development process, and thus no similarities were maintained.
Reviewer 3 Report
Very interesting and well written manuscript, interesting results
Author Response
The authors Tokumaru et al. investigate whether the adenoma-cancer sequence in colon tissue can be recapitulated in organoids. While the answer to this question has implications in investigating mechanisms for early cancer detection and intervention, the presented work has several major limitations. The number of adenoma and CRC organoid lines is too small to draw conclusions with statistical power on the differential expression between adenoma and CRC organoids (n=5 and 7, respectively). The results interpretations need to be revised. The novelty of the findings is limited.
Matano et al. previously generated the dataset from organoids and published their comparison with a different public microarray dataset from adenoma and carcinoma patients than the one used in the reviewed manuscript. They confirmed with the same method (GSEA) that gene expression alterations are retained in organoid culture, albeit not going into specific pathways.
In line 135, ‘These results suggest that cancer organoids derived from adenoma organoids may not proliferate as much as suggested by its expression of cell proliferation-related genes as compared to cancer tissue derived from adenoma tissue’. This sentence misleadingly suggests that the cancer organoids are derived from adenomas. Matano et al. describe that adenomas and carcinoma samples were each isolated from different patients. Similarly, this sentence also suggests that the adenomas from the human tissue are derived from CRC, while most of the adenomas in the dataset do not have a matched primary tumour (from the same patient). Even when the adenomas and primary tumours come from the same patient, there is no evidence that they are related phylogenetically. The authors use ‘cancer organoids derived from adenoma organoids’ throughout the text.
The same misinterpretation is in line 160 ‘These results suggest that organoids generated from non-cancer tissue, such as an adenoma, may have transformed to the point that its transcriptomic profile no longer resembles the profile of cancer tissues’. The cancer organoids were not generated from non-cancer tissue as indicated in the original study of the dataset (Matano et al.). Most importantly, the cancer organoids cannot be directly compared to the cancer tissue, because they do not derive from the same patient. Therefore, it cannot be inferred whether or not the organoids resemble the profile of cancer tissue. The results only show that there is no difference between adenoma and CRC organoids. Finally, these conclusions can be disputed considering that the number of samples (n=5) used for GSEA analysis is small. The authors need to revise the interpretation of the results.
The authors present only a selection of proliferation-associated cancer hallmarks. It would be necessary to give an overview with all cancer hallmarks (related to inflammation, metabolism, development, etc.) compared between adenomas and carcinomas in tissue and organoids.
It is well known that the organoid development does not reflect the immune representation by tissue-resident immune cells or vasculature formation. Therefore, the observation that angiogenesis enrichment scores, stromal scores, and immune infiltration do not differ between adenoma and CRC organoids reflects the known caveats of the model system. Moreover, the stromal heterogeneity can be significant even between two adenomas from the same patient/organoid. Similarly, the composition of the immune cell compartment in a tumour tissue can differ largely in spatially distant parts of the tumour. To capture this heterogeneity with sufficient statistical power, a higher number of organoids is required. Moreover, the variability in immune infiltration is known to be heterogeneous among patients. Matched gene expression from organoids and tissue from the same patient is necessary to confirm whether the TME is recapitulated and to then compare between adenoma and CRC organoids.
Author Response
Author’s Point-by-point Response to the Reviewer’s Comments
Manuscript ID: cells-1081017
Type of manuscript: Original Article
Title: Do organoids mimic the colon adenoma-carcinoma sequence?
Reviewer 4
Comment 1:
The authors Tokumaru et al. investigate whether the adenoma-cancer sequence in colon tissue can be recapitulated in organoids. While the answer to this question has implications in investigating mechanisms for early cancer detection and intervention, the presented work has several major limitations. The number of adenoma and CRC organoid lines is too small to draw conclusions with statistical power on the differential expression between adenoma and CRC organoids (n=5 and 7, respectively). The results interpretations need to be revised. The novelty of the findings is limited.
Response 1:
We would like to thank Reviewer 4 for taking his/her time and putting his/her effort to review our manuscript. The constructive comments helped us improve our manuscript significantly. Although we agree with the Reviewer that the number of adenoma and CRC organoids is small, we do believe that the statistical significance is real when the difference exists despite the small sample size. We have added this comment in the Discussion section as below (Line 356 - 358).
Although we believe that the statistical significance is real when the difference exists despite the small sample size, we may not be encompassing many findings due to the small sample size of organoids.
Comment 2:
Matano et al. previously generated the dataset from organoids and published their comparison with a different public microarray dataset from adenoma and carcinoma patients than the one used in the reviewed manuscript. They confirmed with the same method (GSEA) that gene expression alterations are retained in organoid culture, albeit not going into specific pathways.
Response 2:
We agree with Reviewer 4 that it is important to note that cancer organoids retain the same gene alteration of the original cancer tissue from which the organoids are derived. On the other hand, we believe that the novelty of the current study is that we compared adenoma to CRC derived organoids, as well as adenoma and CRS tissue samples, as opposed to previous studies that compared the organoids to its derived original tumor. We admit that our description in the original manuscript version was not clear on this point, thus the Discussion section was re-written as below (line 307 - 321).
Multiple studies have investigated the gene expression alterations between tumors and its derivative organoids. Matano et. al. reported that cancer organoids maintained the aggressive biology of cancer tissues by expressing CRC-specific genes [12]. Fujii et. al. further demonstrated that CRC organoids preserved the same gene signatures as well as gene alterations of the original CRC tissue by GSEA [31]. In agreement, Weeber et. al. reported that cancer organoids demonstrated similar genetic profiles compared to the original tumor [13]. Van de Wetering et. al. reported the establishment of an organoid bank in which organoids possessed the genetic alterations representative of CRC, such as alterations in APC, TP53, PIK3CA and KRAS [14]. In the current study, the transcriptome of organoids was not compared to the tumor it was derived from. The novelty of our study is the comparison between adenoma and CRC derived organoids, as well as between adenoma and CRS tissue samples, and in the investigation of the differences. We found that cancer or-ganoids enriched the cell proliferation-related gene sets more than adenoma organoids. The same relationship is present in the comparison between cancer tissues and adenoma tissues.
Comment 3:
In line 135, ‘These results suggest that cancer organoids derived from adenoma organoids may not proliferate as much as suggested by its expression of cell proliferation-related genes as compared to cancer tissue derived from adenoma tissue’. This sentence misleadingly suggests that the cancer organoids are derived from adenomas. Matano et al. describe that adenomas and carcinoma samples were each isolated from different patients. Similarly, this sentence also suggests that the adenomas from the human tissue are derived from CRC, while most of the adenomas in the dataset do not have a matched primary tumour (from the same patient). Even when the adenomas and primary tumours come from the same patient, there is no evidence that they are related phylogenetically. The authors use ‘cancer organoids derived from adenoma organoids’ throughout the text.
Response 3:
We apologize for our poor choice of word “derived” which was misleading and confusing throughout the text. We totally agree with Reviewer 4 that Matano et al. clearly stated that adenomas and carcinoma samples were isolated from different patients in their manuscript (PMID: 25706875). Our intention of the description “cancer organoids “derived” from adenoma organoids” was to emphasize the concept of adenoma-carcinoma sequence that cancer has developed from adenoma; however, clearly our description did not reflect our intention and mislead the readers that cancer organoids were derived from adenoma organoids. To this end, we have revised the manuscript as follows.
Line 145 - 147
These results suggest that cancer organoids may be as proliferative as compared to adenoma organoids, as cancer tissue is as proliferative as adenoma tissue, suggested by the enrichment of cell proliferation-related gene sets.
Line 182 - 183
However, cancer organoids did not enrich either of these gene sets (Figure 3B).
Line 187 - 188
On the contrary, stromal cells were not highly infiltrated in cancer organoids.
Comment 4:
The same misinterpretation is in line 160 ‘These results suggest that organoids generated from non-cancer tissue, such as an adenoma, may have transformed to the point that its transcriptomic profile no longer resembles the profile of cancer tissues’. The cancer organoids were not generated from non-cancer tissue as indicated in the original study of the dataset (Matano et al.). Most importantly, the cancer organoids cannot be directly compared to the cancer tissue, because they do not derive from the same patient. Therefore, it cannot be inferred whether or not the organoids resemble the profile of cancer tissue. The results only show that there is no difference between adenoma and CRC organoids. Finally, these conclusions can be disputed considering that the number of samples (n=5) used for GSEA analysis is small. The authors need to revise the interpretation of the results.
Response 4:
We apologize for the confusion. We totally agree with Reviewer 4 that Matano et al established cancer organoids from cancer tissue and not from non-cancer tissue. We also agree that whether or not the organoids resemble the profile of cancer tissue cannot be inferred given that those samples were not obtained from same patient. We revised the following sentence in the Result section (Line 168 - 170).
These results suggest that cancer organoids do not enrich many of the adenoma-carcinoma sequence related gene sets as cancer tissue.
Comment 5:
The authors present only a selection of proliferation-associated cancer hallmarks. It would be necessary to give an overview with all cancer hallmarks (related to inflammation, metabolism, development, etc.) compared between adenomas and carcinomas in tissue and organoids.
Response 5:
We would like to thank Reviewer 4 for this constructive comment. We totally agree with the Reviewer that it will be informative to the readers to demonstrate all the significantly enriched gene sets in cancer tissues or cancer organoids. To this end, we added Table S1 and Table S2 as follows.
|
Table S1: Gene sets enriched in colorectal cancer tissue (GSE41258) |
|||
|
NAME |
NES |
FDR |
Category |
|
HALLMARK_EPITHELIAL_MESENCHYMAL_ TRANSITION |
-2.02 |
0.011 |
Development |
|
HALLMARK_ANGIOGENESIS |
-1.93 |
0.024 |
Development |
|
HALLMARK_APICAL_JUNCTION |
-1.90 |
0.024 |
Cellular component |
|
HALLMARK_UV_RESPONSE_UP |
-1.84 |
0.032 |
DNA damage |
|
HALLMARK_MYOGENESIS |
-1.70 |
0.105 |
Development |
|
HALLMARK_IL2_STAT5_SIGNALING |
-1.62 |
0.170 |
Signaling |
|
HALLMARK_MITOTIC_SPINDLE |
-1.60 |
0.167 |
Proliferation |
|
HALLMARK_HEDGEHOG_SIGNALING |
-1.59 |
0.161 |
Signaling |
|
HALLMARK_MTORC1_SIGNALING |
-1.58 |
0.150 |
Signaling |
|
HALLMARK_UV_RESPONSE_DN |
-1.58 |
0.138 |
DNA damage |
|
HALLMARK_KRAS_SIGNALING_UP |
-1.57 |
0.135 |
Signaling |
|
HALLMARK_TNFA_SIGNALING_VIA_NFKB |
-1.57 |
0.127 |
Signaling |
|
HALLMARK_G2M_CHECKPOINT |
-1.56 |
0.121 |
Proliferation |
|
HALLMARK_INFLAMMATORY_RESPONSE |
-1.55 |
0.119 |
Immune |
|
HALLMARK_APOPTOSIS |
-1.53 |
0.125 |
Pathway |
|
HALLMARK_COMPLEMENT |
-1.52 |
0.130 |
Immune |
|
HALLMARK_HYPOXIA |
-1.52 |
0.124 |
Pathway |
|
HALLMARK_APICAL_SURFACE |
-1.50 |
0.126 |
Immune |
|
HALLMARK_INTERFERON_GAMMA_RESPONSE |
-1.48 |
0.135 |
Immune |
|
HALLMARK_E2F_TARGETS |
-1.47 |
0.133 |
Proliferation |
|
HALLMARK_COAGULATION |
-1.46 |
0.138 |
Immune |
|
HALLMARK_IL6_JAK_STAT3_SIGNALING |
-1.44 |
0.144 |
Immune |
|
HALLMARK_ALLOGRAFT_REJECTION |
-1.43 |
0.145 |
Immune |
|
HALLMARK_INTERFERON_ALPHA_RESPONSE |
-1.42 |
0.150 |
Immune |
|
HALLMARK_UNFOLDED_PROTEIN_RESPONSE |
-1.39 |
0.166 |
Pathway |
|
HALLMARK_WNT_BETA_CATENIN_SIGNALING |
-1.36 |
0.184 |
Signaling |
|
HALLMARK_TGF_BETA_SIGNALING |
-1.30 |
0.234 |
Signaling |
|
HALLMARK_ESTROGEN_RESPONSE_EARLY |
-1.29 |
0.237 |
Signaling |
|
Table S2: Gene sets enriched in colorectal cancer organoids (GSE57965) |
|||
|
NAME |
NES |
FDR |
Category |
|
HALLMARK_MYC_TARGETS_V2 |
-1.91 |
0.002 |
Proliferation |
|
HALLMARK_UNFOLDED_PROTEIN_RESPONSE |
-1.64 |
0.120 |
Pathway |
|
HALLMARK_UV_RESPONSE_UP |
-1.60 |
0.113 |
DNA damage |
|
HALLMARK_E2F_TARGETS |
-1.58 |
0.100 |
Proliferation |
|
HALLMARK_MYC_TARGETS_V1 |
-1.52 |
0.118 |
Proliferation |
|
HALLMARK_G2M_CHECKPOINT |
-1.48 |
0.132 |
Proliferation |
|
HALLMARK_MTORC1_SIGNALING |
-1.38 |
0.197 |
Signaling |
Comment 6:
It is well known that the organoid development does not reflect the immune representation by tissue-resident immune cells or vasculature formation. Therefore, the observation that angiogenesis enrichment scores, stromal scores, and immune infiltration do not differ between adenoma and CRC organoids reflects the known caveats of the model system.
Response 6:
We totally agree with Reviewer 4 that it is well known that organoid development does not reflect the immune representation by tissue-resident immune cells or vasculature formation of the original cancer tissue that it was derived from. Indeed, this is a known caveat of this model system. On the other hand, the novelty of the current study is that it investigated the difference in stromal cells and vasculature formation between CRC organoid and adenoma organoid. This was of interest because theoretically if tissue-resident immune cells and vascular cells were lost in exact same ratio by generation of organoids, the difference in those cells between CRC and adenoma organoids should be the same as CRC and adenoma tissues. As a result, we found that that was not the case; however, we do not think our result is a mere reflection of the known caveats of the model. To make this statement clear, we revised a part of discussion as follow (Line 327 – 334).
A limitation of organoids is that they do not proportionally reflect the composition of the tumor microenvironment as represented by tissue-resident immune cells or vasculature promoting cells of the original cancer tissue [31,32]. The novelty of the current study is that it investigates the difference in stromal and vasculature formation between CRC organoids and adenoma organoids. The results indicated that there was lack of fidelity in the proportional loss of tissue-resident and vasculature promoting cells between CRC and adenoma organoids versus tissues. This implies that immune and vascular cells are lost during the organoid development process, and thus no similarities were maintained.
Comment 7:
Moreover, the stromal heterogeneity can be significant even between two adenomas from the same patient/organoid. Similarly, the composition of the immune cell compartment in a tumour tissue can differ largely in spatially distant parts of the tumour. To capture this heterogeneity with sufficient statistical power, a higher number of organoids is required. Moreover, the variability in immune infiltration is known to be heterogeneous among patients. Matched gene expression from organoids and tissue from the same patient is necessary to confirm whether the TME is recapitulated and to then compare between adenoma and CRC organoids.
Response 7:
We totally agree with Reviewer 4 that there is a significant spatial heterogeneity in infiltrating immune cell compositions within a bulk tumor let alone among the different patients. As suggested, sufficient statistical power is required to overcome this issue. Also, ideally the organoid and tissue needs to be obtained from same patient to confirm that the gene expression of organoid and tissue is almost identical before the comparison of those samples. However, it is very hard to figure out the ideal number to reflect the heterogeneity of tumor immune microenvironment. Also, unfortunately, at this moment, we do not have access to the data of ideal number of organoids and matched tissue samples to reflect the heterogeneity of tumor immune microenvironment. To this end, we revised the limitation part of discussion as follows (Line 356 – 365).
The current study has obvious limitations. One of the limitations stems from our inability to able to compare tissue samples and organoid samples in the same population. This was due to the lack of access to the cohort which has both tissue and organoid transcriptomic data. In addition, our result may not reflect the spatial difference within the bulk tumor and the heterogeneity among the patients due to the small number of organoid samples. Although we believe that the statistical significance is real when the difference exists despite the small sample size, we may not be encompassing many findings due to the small sample size of organoids. Further investigation that compares organoid and tissue samples directly is warranted and would make our findings more intriguing and convincing.
Round 2
Reviewer 4 Report
The authors have addressed some of my comments by disclosing the limitations of the study in the text. However, the major concerns that the conclusions were drawn with insufficient statistical power and the novelty was limited, were not addressed. To strengthen the conclusions and methodological approach of the manuscript, the authors could have included other publicly available datasets, such as some of the listed studies in Response 2.
While the significant statistical difference may be ‘real’, it should be considered that the threshold used in the manuscript FDR<0.25 indicates that we expect 25% false positives (1 in 4).
Optionally, in addition to performing GSEA analysis to compare two groups, the authors could have considered single-sample GSEA, to avoid the sample size limitations in comparison between two groups.
It should be noted that the authors argue that ‘the novelty of the current study is that we compared adenoma to CRC derived organoids, as well as adenoma and CRS tissue samples, as opposed to previous studies that compared the organoids to its derived original tumor’, in Response 2. Of note, the original study by Matano et al. also compared adenoma to carcinoma separately in tissue and organoids, using the same approach on all differentially expressed genes (the organoids were not derived from the compared tissue).
Author Response
Author’s Point-by-point Response to the Reviewer’s Comments
Manuscript ID: cells-1081017
Type of manuscript: Original Article
Title: Do organoids mimic the colon adenoma-carcinoma sequence?
Reviewer 4
Comment 1:
The authors have addressed some of my comments by disclosing the limitations of the study in the text. However, the major concerns that the conclusions were drawn with insufficient statistical power and the novelty was limited, were not addressed. To strengthen the conclusions and methodological approach of the manuscript, the authors could have included other publicly available datasets, such as some of the listed studies in Response 2.
Response 1:
We would like to thank Reviewer 4 for his/her rapid response and spending time and effort in reviewing our manuscript.
We agree with Reviewer 4 that the original cohort is small and that was a weakness. To this end, we also agree adding analyses using the cohort by Fujii et al (GSE74843) strengthened the conclusion and methodological approach of the manuscript. New data by those analyses and modified Materials and Methods, Results, and Discussion sections are as follows.
Materials and Methods
The transcriptomic and clinical data of colorectal cancers (CRC) and adenomas were obtained from three cohorts; one cohort of tissues (GSE41258) [16] and two cohorts of organoids (GSE57965 [12] and GSE74843) [29].
GSE74843 holds a total of 58 samples which includes: 10 adenoma organoids, 38 CRC organoids, 7 normal colorectal mucosa organoids, and 3 serrate organoids. Among those, we utilized a total of 48 samples (10 adenoma organoids and 38 CRC organoids).
Results
3.2. Among the pathways in colon adenoma-carcinoma sequence, only the MTORC1 gene set was enriched in cancer organoids
On the other hand, only the MTORC1 pathway was enriched in cancer organoids in GSE57965 (Figure 2B, Table S2; NES = -1.38 FDR = 0.197). TGF-β pathway was enriched in adenoma organoids (Figure 2B, NES = -1.68, FDR = 0.152), which was consistent in GSE74843 (Figure 2C; NES = -1.77 FDR = 0.182).
Figure 2. GSEA of gene sets associated with adenoma-carcinoma sequence. (A) Analysis of tissue sample (GSE41258). (B) Analysis of organoid sample (GSE57965). (C) Analysis of organoid sample (GSE74843). Statistical significance was defined as false discovery rate (FDR) < 0.25. CRC, colorectal cancer; FDR, false discovery rate.
3.3. Only cancer tissues enriched tumor immune microenvironment (TME)-related gene sets and correlated with a higher infiltration of stromal cells when compared to adenoma tissues.
In GSE74843, only fibroblasts were highly infiltrated in adenoma organoids and this trend was consistent with the Stromal Score (Supplementary Figure S2).
Supplementary Figure S2. GSEA of tumor immune microenvironment (TME)-related gene sets, the infiltration of stromal cells and the comparison of Stroma Scores. (A) GSEA of tissue sample (GSE74843). (B) Infiltration of stromal cells and comparison of Stroma Scores in organoid samples (GSE74843). Tukey-type boxplots demonstrate the median as well as interquartile level values. Statistical significance was defined as false discovery rate (FDR) < 0.25. Ade, adenoma; CRC, colorectal cancer
3.4. Cancer tissue, but not cancer organoids, enriched immune response-related gene sets
Cancer organoids did not enrich any of the immune response-related gene sets compared to adenoma organoids in either GSE57965 (Figure 4B) or GSE74843 (Supplementary Figure S3). These results suggest that immune activity in cancer organoids is no different from adenoma organoids.
Supplementary Figure S3. GSEA of gene sets associated with immune response. Analysis of tissue sample (GSE74843).
3.6. B cells and T cells were overall less infiltrated and Macrophages were more infiltrated in cancer tissue compared to adenoma tissue, whereas there was no difference in organoids
Only CD4 memory resting T cells were significantly lower in cancer organoids as compared to adenoma organoids in both GSE57965 (Figure 6A; p = 0.012) and GSE74843 (Supplementary Figure S4A; p = 0.046) whereas CD4 memory activated cells were significantly higher in only GSE57965 (Figure 6A; p = 0.049).
Cancer organoids demonstrated consistent higher infiltration of M0 in GSE74843 (Supplementary Figure S4B; p = 0.015), whereas activated DCs were consistently infiltrated in cancer organoids of GSE57965 (Figure 6B; p = 0.005).
There was no difference between cancer and adenoma tissue or organoids in infiltration of NK cells and eosinophils (Supplementary Figure S5B and S5C). In GSE74843, resting mast cells demonstrated higher infiltration in cancer organoids than adenoma organoids, whereas activated mast cells demonstrated the opposite result (Supplementary Figure S5C). Neutrophils were higher in cancer tissue, but not in cancer organoids (Supplementary Figure S5).
Supplementary Figure S4. Comparison of the infiltration of immune cells of GSE74843. (A) Analysis of T cells. (B) Analysis of Monocytes. Tukey-type boxplots demonstrate the median as well as interquartile level values. Ade, adenoma; CRC, colorectal cancer
Supplementary Figure S5. Comparison of the infiltration of immune cells (B cells, NK cells, and Granulocytes). (A) Analysis of B cells. (B) Analysis of NK cells. (C) Analysis of Granulocytes Tukey-type boxplots demonstrate the median as well as interquartile level values. Ade, adenoma; CRC, colorectal cancer; NK cell, natural killer cell
3.7. There was no difference in the infiltration of stem cells and progenitor cells between adenoma and cancer organoids
On the contrary, cancer organoids demonstrated no difference in either type of stem cell as compared to adenoma organoids in both GSE57965 and GSE74843 (Figure 7B, Supplementary Figure S6A).
Only megakaryocyte-erythroid progenitor cells were highly infiltrated in cancer organoids of GSE74843 (Supplementary Figure S6B) whereas there was no difference in progenitor cells between cancer and adenoma organoids in GSE57965 (Figure 7D).
Supplementary Figure S6. Comparison of infiltration of stem cells and progenitor cells between adenoma and CRC tissue and organoid. (A) Infiltration of stem cells in tissue samples (GSE74843). (B) Infiltration of progenitor cells in tissue samples (GSE74843).
Discussion
Only the MTORC1 pathway, among all the pathways involved in adenoma-carcinoma sequence (WNT beta catenin, KRAS signaling up, MTORC1, and TGF-β pathways), was enriched in cancer organoids of GSE 57965. In comparison, all of the pathways were enriched in cancer tissue.
Comment 2:
While the significant statistical difference may be ‘real’, it should be considered that the threshold used in the manuscript FDR<0.25 indicates that we expect 25% false positives (1 in 4). Optionally, in addition to performing GSEA analysis to compare two groups, the authors could have considered single-sample GSEA, to avoid the sample size limitations in comparison between two groups.
Response 2:
We would like to thank Reviewer 4 for his/her constructive comment. We totally agree with the Reviewer’s comment on FDR, thus we performed single-sample GSEA for the cell proliferation-related gene sets as suggested. Also, we modified Result section as follows.
3.1. Cell proliferation-related gene sets were enriched in colorectal cancer (CRC) in both tissue and organoid cohorts
To overcome the sample size limitation, we performed single-sample GSEA on GSE57965. The results of single-sample GSEA were similar to that of GSEA. The gene sets that enriched to CRC tissue, such as E2F Targets, G2M Checkpoint, Myc Targets V1 and V2, were also significantly enriched in CRC organoids, which had higher scores of single-sample GSEA than adenoma organoids (Figure 1E; p = 0.024, p = 0.037, p = 0.036, and p < 0.001 respectively). Also, we performed another cohort, GSE74843, which includes adenoma organoids and CRC organoids. Interestingly, In GSE74843, CRC organoids enriched Myc Target V2 gene set which was further was validate with single-sample GSEA. CRC organoids demonstrated higher single-sample GSEA score than adenoma organoids in Myc Target V2 gene set (Supplementary Figure S1).
Figure 1. Gene set enrichment analysis (GSEA) of cell proliferation-related gene sets and analysis of MKI67 expression. (A) GSEA of adenoma vs CRC tissue in GSE41258. (B) The MKI67 expression levels of adenoma and CRC in tissue. (C) GSEA of adenoma vs CRC organoids in GSE57965. (D) The comparison of MKI67 expression levels between adenoma and CRC in organoids. Tukey-type boxplots demonstrate the median as well as interquartile level values. (E) Single-sample GSEA of GSE57965. Statistical significance was defined as false discovery rate (FDR) < 0.25. Ade, adenoma; CRC, colorectal cancer; FDR, false discovery rate.
Supplementary Figure S1. Gene set enrichment analysis (GSEA) of cell proliferation-related gene sets and analysis of MKI67 expression. (A) GSEA of adenoma vs CRC tissue in GSE74843. (B) The MKI67 expression levels in adenoma and CRC in tissue. (C) Single-sample GSEA of GSE74843. Statistical significance was defined as false discovery rate (FDR) < 0.25. Ade, adenoma; CRC, colorectal cancer; FDR, false discovery rate.
Comment 3:
It should be noted that the authors argue that ‘the novelty of the current study is that we compared adenoma to CRC derived organoids, as well as adenoma and CRS tissue samples, as opposed to previous studies that compared the organoids to its derived original tumor’, in Response 2. Of note, the original study by Matano et al. also compared adenoma to carcinoma separately in tissue and organoids, using the same approach on all differentially expressed genes (the organoids were not derived from the compared tissue).
Response 3:
We agree with the Reviewer 4 that we should give a proper credit to the original study by Matano et al, which has compared the adenoma to cancer organoids as well as tissues. To this end, our statement in Response 2 may be overstating. We have modified the sentence in Discussion as follows.
In the original study, Matano et al. compared adenoma to CRC in tissues and organoids using differentially expressed genes to analyze whether the organoids demonstrated the gene characteristics of original patients [12]. The novelty of our approach is that we used Hallmark gene sets of Molecular Signature Database collections in GSEA, which allows a bird-eye-view of biology as demonstrated by hallmarks of cancer, such as cell proliferation, inflammation, and metabolism.